# Improvement of the Diagnostic Performance of Facial Neuritis Using Contrast-Enhanced 3D T1 Black-Blood Imaging: Comparison with Contrast-Enhanced 3D T1-Spoiled Gradient-Echo Imaging

**DOI:** 10.3390/jcm10091850

**Published:** 2021-04-24

**Authors:** Seun-Ah Lee, Sang-Won Jo, Suk-Ki Chang, Ki-Han Kwon

**Affiliations:** 1Department of Radiology, Dongtan Sacred Heart Hospital, Hallym University Medical Center, 7, Keunjaebong-gil, Hwaseong-si 18450, Gyeonggi-do, Korea; leesa@hallym.or.kr (S.-A.L.); csk1270@hallym.or.kr (S.-K.C.); 2Department of Neurology, Dongtan Sacred Heart Hospital, Hallym University Medical Center, 7, Keunjaebong-gil, Hwaseong-si 18450, Gyeonggi-do, Korea; neurokkh@hallym.or.kr

**Keywords:** 3D T1 black-blood imaging, facial neuritis, diagnostic performance, comparison study, 3D T1-spoiled gradient-echo imaging

## Abstract

This study aims to investigate the diagnostic ability of the contrast-enhanced 3D T1 black-blood fast spin-echo (T1 BB-FSE) sequence compared with the contrast-enhanced 3D T1-spoiled gradient-echo (CE-GRE) sequence in patients with facial neuritis. Forty-five patients with facial neuritis who underwent temporal bone MR imaging, including T1 BB-FSE and CE-GRE imaging, were examined. Two reviewers independently assessed the T1 BB-FSE and CE-GRE images in terms of diagnostic performance, and qualitative (diagnostic confidence and visual asymmetric enhancement) and quantitative analysis (contrast-enhancing lesion extent of the canalicular segment of the affected facial nerve (LEC) and the affected side-to-normal signal intensity ratio (rSI)). The AUCs of each reviewer, and the sensitivity and accuracy of T1 BB-FSE were significantly superior to those of CE-GRE (*p* < 0.05). Regarding diagnostic confidence and visual asymmetric enhancement, T1 BB-FSE tended to be rated greater than CE-GRE (*p* < 0.05). Additionally, in quantitative analysis, LEC and rSI of the canalicular segment on T1 BB-FSE were larger than those on CE-GRE (*p* < 0.05). The T1 BB-FSE sequence was significantly superior to the CE-GRE sequence, with more conspicuous lesion visualization in terms of both qualitative and quantitative aspects in patients with facial neuritis.

## 1. Introduction

Facial neuritis (Bell’s palsy) is the most common cause of unilateral peripheral facial nerve palsy [1]. Patients with facial neuritis usually present rapid unilateral facial nerve weakness/paralysis without a clear cause [2]. Although facial neuritis is considered idiopathic, the main hypothesized cause is herpes virus reactivation, supported by the detection of herpes virus DNA in the endoneural fluids of patients with facial neuritis [3,4]. Inflammation, myelin breakdown, and hemorrhage of the facial nerve, which are caused by viral infection in the narrow facial nerve canal of the temporal bone, lead to edema, compression, and ischemia [5,6,7]. This ischemia can result in degeneration and axonal destruction of the facial nerve [5,8,9].

Facial neuritis is diagnosed primarily on the basis of characteristic clinical symptoms, physical examination, magnetic resonance imaging (MRI), and electrodiagnostic testing [10,11]. Electrodiagnostic testing can also offer prognostic information about the outcomes in patients with facial neuritis [10,11].

However, electrodiagnostic testing has limitations in diagnosing the early stages of facial neuritis. For example, electroneurography cannot detect abnormalities for approximately 72 h until Wallerian degeneration from the intratemporal area reaches the extratemporal area, the distal portion of the stylomastoid foramen. Additionally, electromyography cannot detect signs of muscle degeneration due to facial nerve palsy within 10–14 days [10].

In contrast, MRI can provide significant diagnostic information on facial neuritis within one week of symptom onset, which is an important time for prognostic and recovery rate evaluation [11,12,13,14,15]. Therefore, in initial evaluations of facial neuritis, MRI can play important roles not only in the exclusion of other diseases but also in the establishment of a rapid diagnosis and the selection of an appropriate treatment.

There have been many studies focused on improving the diagnostic ability of facial neuritis using MRI over the decades. It has been revealed that the manifestation of facial neuritis on MRI is an abnormally intense enhancement in the distal intracanalicular segment and labyrinthine segment that does not appear in normal facial nerves [16,17,18]. Thus, a comparison with the symptom-free healthy side in contrast-enhanced MRI helps to distinguish it from intense enhancement of the affected side, which shows symptoms [19].

However, the facial nerve, similar to other cranial nerves, is surrounded by several layers of connective tissue represented by the endoneurium, perineurium, and epineurium and has a relatively abundant arteriovenous plexus in the connective tissue. Therefore, enhancement of the facial nerve may be normal due to pooling of the contrast agent in the blood vessels of the perineurium, but it can also be abnormal, such as when the contrast caused by destruction of the blood–nerve barrier is enhanced [20]. These findings mean that distinguishing a slight abnormal enhancement caused by disruption of the blood–peripheral nerve barrier from normal enhancement is likely to be a hindrance.

The recently developed 3D T1 black-blood fast spin-echo imaging method suppresses signals from blood flow and provides an improved contrast-to-noise ratio (CNR) compared with other sequences [21]. Consequently, some recent studies using contrast-enhanced 3D T1 black-blood fast spin-echo imaging (T1 BB-FSE) have reported improved small lesion detection rates of variable intra- and extracranial diseases [22,23,24,25,26,27,28]. However, to our best knowledge, no study has examined the diagnostic accuracy of the T1 BB-FSE sequence for facial neuritis.

Therefore, this study aimed to investigate whether the T1 BB-FSE sequence has better diagnostic ability than the contrast-enhanced 3D T1-spoiled gradient-echo with fat suppression (CE-GRE) sequence in patients clinically diagnosed with facial neuritis.

## 2. Materials and Method

This retrospective study was approved by the institutional review board of our institution, and the requirement for informed consent was waived.

### 2.1. Patients

Initially, we enrolled 63 consecutive patients with acute unilateral facial palsy who underwent temporal bone MRI within one month of symptom onset from 1 November 2019 to 31 December 2020. Of these, 13 patients were excluded for the following reasons: (1) brain stem (pons or medulla oblongata) infarction was detected in MRI (*n* = 5), (2) the patients were diagnosed with Ramsay-hunt syndrome (*n* = 5), (3) the patients had intracranial tumors on the symptomatic side (*n* = 1), (4) a patient showed erosion of the fallopian canal caused by cholesteatoma on temporal bone CT (*n* = 1), and (5) a patient was diagnosed with parotid gland adenocarcinoma by MRI and excisional biopsy (*n* = 1).

Of the remained 50 consecutive patients with acute facial neuritis, two were excluded because of severe dental prosthesis artifacts and one was excluded because he was younger than 18 years of age. Two patients were excluded because of the symptom onset occurring more than one month before MRI. Finally, 45 patients were included in our study. As facial neuritis is a diagnosis of exclusion, all 45 patients were finally diagnosed with unilateral facial neuritis by thorough medical history assessment; clinical examination, including otoscopy; meticulous cranial nerve assessment; and follow-up [2]. The clinical characteristics of the study population are summarized in Table 1.

### 2.2. MRI Protocol

MR scans, which included the CE-GRE and T1 BB-FSE sequences, were conducted using one of two 3T scanners (Skyra or Verio; Siemens, Erlangen, Germany) with a 32-channel head coil. The CE-GRE sequence and T1 BB-FSE sequences were obtained after intravenous administration of gadobutrol (Gadovist; Bayer Schering Pharma, Berlin, Germany) at a dose of 0.1 mmol/kg of body weight. The scanning order was constant in this study, and T1 BB-FSE (time delay: 7 min) was scanned after CE-GRE (time delay: 2 min). MR imaging scans were performed in 21 and 24 patients using Skyra and Verio scanners, respectively. The detailed MRI parameters of each sequence are depicted in Appendix A.

### 2.3. MR Image Analysis

Two neuroradiologists (reviewers) who were unaware of the patient’s clinical history, including the affected side and proportion of facial neuritis, independently evaluated the 65 temporal bone MR imaging data sets (45 of the patients with facial neuritis and 20 of the normal controls) to compare the diagnostic ability of both MR sequences. Before MRI analysis, we randomly sampled 10 cases (3 normal cases and 7 facial neuritis cases) for a training interpretation session. After the training session, the two reviewers individually interpreted the CE-GRE sequences first and then the T1 BB-FSE sequences in at least two-week intervals. Each patient in the two MRI data sets was randomly extracted and analyzed in different image analysis sessions. The following three-point rating scale was used to evaluate the diagnostic confidence based on asymmetric contrast enhancement of the facial nerve (canalicular and/or labyrinthine segment): 0, normal; 1, suspicious finding; and 2, definite finding. Furthermore, the following four-point rating scale was used to assess the visual degree of contrast enhancement of the three segments (canalicular, labyrinthine, and anterior genu) of the affected facial nerves (if any): 0, none; 1, faint; 2, moderate; and 3, intense. Regarding the anterior genu segment, we also evaluated the visual contrast enhancement degree of the contralateral anterior genu segment with no symptoms. The visual contrast enhancement degree of the affected side was subtracted from that of the normal side. The driven difference was defined as the asymmetric enhancement degree of the anterior genu segment.

### 2.4. Quantitative Analysis

First, the contrast-enhancing lesion extent (length and width) of the canalicular segment of the abnormal facial nerve (LEC) was measured using the ruler tool of PACS by one of two neuroradiologists. Second, the signal intensity (SI) was estimated using regions of interest (ROIs) at the three segments (canalicular, labyrinthine, and anterior genu) of the bilateral facial nerves by the same neuroradiologist. The same neuroradiologist measured the LEC and SI of each segment three times to increase the reliability of the quantitative values. These quantitative measurements were conducted on the 90 facial nerves of all 45 patients with facial neuritis. The derived mean value for quantitative analysis was used for statistical analysis. The mean value of SI was used to defined the relative signal intensity increase (rSI: affected side-to-normal signal intensity ratio) for the two MR sequences.

### 2.5. Statistical Analysis

We used receiver operating characteristic (ROC) curve analysis to assess the diagnostic performances of the two MR sequences in diagnosing facial neuritis. For each reviewer, we used DeLong’s test to compare the area under the curve (AUC) for the two MRI data sets. The diagnostic accuracy (sensitivity, specificity, and accuracy) of the two investigators was calculated using 2 × 2 tables and compared between each MR imaging sequence using McNemar’s test. The Wilcoxon signed-rank test was used to compare the diagnostic confidence and visual asymmetric enhancement between the two MR imaging sequences for the two reviewers. The interrater agreement was analyzed using the percentage of agreement and the intraclass correlation coefficient (ICC). We also evaluated the difference in the quantitative lesion extent and rSI between the two MR imaging sequences by using the Mann–Whitney test. The interrater agreement of qualitative measures and intra-rater agreement of the quantitative measures were evaluated by computing the intraclass correlation coefficient (ICC). All statistical analyses were performed using a statistical software program (Version 19.6.4 for Windows; MedCalc, Mariakerke, Belgium). A *p* value < 0.05 was accepted as statistically significant.

## 3. Results

### 3.1. Diagnostic Performance of the MR Imaging Sequences

The ROC curve analysis showed that the diagnostic accuracy of T1 BB-FSE was higher than that of CE-GRE for each reviewer. There was a statistically significant difference between the ROC curves of the two MR imaging sequences in DeLong’s test for each reviewer, and the *p*-values were 0.014 and 0.042. The sensitivity (97.2%) and accuracy (96.2%) of T1 BB-FSE were significantly higher than those (sensitivity, 87.8%; accuracy, 83.8%) of CE-GRE (sensitivity, *p* = 0.004; accuracy, *p* < 0.001). The specificity of T1 BB-FSE (92.5%) was higher than that of CE-GRE (75%) and approached statistical significance (*p* = 0.065). The diagnostic performance outcomes, including the ROC curves of the two reviewers, are provided in Figure 1 and Table 2.

### 3.2. Qualitative Analysis of the Facial Neuritis

Regarding the diagnostic confidence in the facial neuritis patient group, a higher proportion of definite findings and a lower proportion of suspicious findings were revealed with the T1 BB-FSE sequence than with the CE-GRE sequence (*p* < 0.001). Regarding the interrater agreement of diagnostic confidence, each MR sequence showed similar high degrees of agreement; CE-GRE showed an ICC value of 0.896 (95% CI: 0.835–0.935), and T1 BB-FSE showed an ICC value of 0.906 (95% CI: 0.851–0.942). Furthermore, visual asymmetric enhancement of the canalicular and labyrinthine segments was rated higher by each reviewer on T1-BB-FSE than on CE-GRE (*p* < 0.005). In contrast, no significant difference was observed between the two sequences of the anterior genu segment in visual asymmetric contrast enhancement analysis. The ICCs of visual asymmetric contrast enhancement of the three segments between the two reviewers for each MR sequence were similar, ranging from good to excellent (range: 0.64–0.78). Further detailed results of the qualitative analysis are shown in Table 3. Representative cases are shown in Figure 2, Figure 3, Figure 4 and Figure 5.

### 3.3. Quantitative Analysis of the LEC and rSI of Facial Nerves

The mean values of the LEC measured 3 times showed significantly longer and wider enhancement in T1 BB-FSE than in CE-GRE. (*p* < 0.001 for length; *p* = 0.026 for width). The mean values of the three measurements of the rSI were significantly higher in T1 BB-FSE than in CE-GRE for the canalicular segment. However, for the labyrinthine segment and anterior genu segment, the mean values of the rSI between the two sequences showed no significant difference. The detailed LEC and rSI values of each segment of the facial nerve are summarized in Table 4. The ICC for the intra-rater reliability of the LEC and rSI of each facial nerve segment, which was measured three times, was 0.91 or above.

## 4. Discussion

In this study, the abnormal enhancement of facial neuritis was more conspicuous on the T1 BB-FSE sequence than on the CE-GRE sequence. Additionally, the rSI of the abnormal facial nerve was higher in the T1 BB-FSE sequence than in the CE-GRE sequence. Among the facial nerve segments, the rSI of the canalicular segment in the T1 BB-FSE sequence was significantly higher than that in the CE-GRE sequence.

The 3D T1-spoiled gradient-echo (SPGR) sequence can use thin slices, which improve the diagnostic ability for facial neuritis and have been commonly used to assess facial nerves [29,30,31]. However, in a previous study that analyzed the enhancement pattern of a normal facial nerve using a 3D T1 SPGR sequence, enhancements of 15% and 5% were reported in the canalicular segment and labyrinthine segment, respectively [29]. Additionally, the SI of the facial nerve was higher in the inversion recovery prepared fast spoiled gradient-echo (IR-FSPGR) sequence before and after contrast enhancement than that in the CE 2D-SE sequence with fat suppression because of the sequence itself [32]. This finding might be caused by the higher spatial resolution of 3D IR-FSPGR than that of the CE 2D-SE sequence with fat suppression. In daily clinical practice, the 3D T1 pre-contrast image is not routinely obtained because of the limited overall acquisition time. Thus, in this clinical setting without 3D T1 pre-contrast images, hyper-SI in the canalicular and labyrinthine segments of the facial nerve observed on GRE images does not necessarily indicate inflammation of the facial nerve. The cause might be contrast pooling in the perineural arteriovenous plexus, which surrounds the normal facial nerve (Figure 4). These findings might explain the low specificity in diagnosing facial neuritis on CE-GRE in this study.

The 3D FSE sequences have the advantage of being less sensitive to susceptibility artifacts than the 3D GRE sequences; thus, better detectability of the normal facial nerve has been reported compared to using the 3D GRE sequence [30,31]. Several studies have investigated the diagnostic performances of facial neuritis using the T1 3D FSE sequence [33,34,35]. In a recent study using a T1 3D FSE sequence (VISTA), a significant difference in the contrast ratio (abnormal enhancing lesion signal intensity/normal nerve signal intensity) compared to the T1 2D turbo spin-echo (TSE) sequence was reported in patients with facial neuritis. However, the sensitivity, specificity, and accuracy between the two sequences did not reach statistical significance in this study [33]. Another previous study reported that the specificity and accuracy of the CE 3D-FLAIR sequence were higher than those of the T1 3D-SPGR sequence in diagnosing facial neuritis [34]. However, the 3D-FLAIR sequence has a longer acquisition time than the conventional T1 3D-SPGR sequence; thus, a potential limitation exists in comparing pre- and postcontrast 3D-FLAIR sequences for diagnosis. Additionally, in another recent study using the CE 3D-FLAIR sequence and CE T1 2D TSE sequence, the qualitative diagnostic performance outcomes of the CE 3D-FLAIR sequences were lower than those of the CE T1 2D TSE sequence in pediatric patients with facial neuritis [35].

Black-blood imaging, which was recently developed for the selective nullification of moving flow, has shown superior diagnostic ability compared to conventional MR sequences in the detection of small brain metastases, leptomeningeal metastases, leptomeningitis, optic neuritis, and contrast-enhancing lesions in multiple sclerosis [22,23,24,25,26,27,28]. The results of our study are in line with those of previous studies using the T1 BB-FSE sequence. The results of our study are theoretically supported by other previous studies that revealed that lesions in the SE sequence were markedly enhanced by gadolinium than those in the GRE sequence [36,37]. In addition, the contrast-to-noise ratio (CNR) is increased by the motion-sensitized driven-equilibrium (MSDE) technique, which is used to create a T1 black-blood image, and thus, lesion detection is improved [38]. Lastly, in the T1 BB-FSE sequence, the blood flow suppression effect is obtained by using a variable flip angle refocusing pulse as well as MSDE [39]. This sequence constrains image blurring from T2 decay over a long echo train and can achieve isotropic resolution in a relatively short scanning time in a clinical setting [40,41].

In our quantitative analysis, although the rSI of all three segments showed higher values on T1 BB-FSE than on CE-GRE, only the canalicular segment reached statistical significance. We speculate that the anterior genu segment, which shows contrast enhancement frequently, even in the normal facial nerve without symptoms, is an explanation of this result [29,32,42]. Additionally, the labyrinthine and anterior genu segments are relatively smaller areas, and susceptibility artifacts from the enclosed bony structures might cause low signal-to-noise ratios (SNRs) and CNRs in these regions on CE-GRE compared with those on T1 BB-FSE [43]. Therefore, we speculate that the SIs of the slight enhancement in these two segments at the asymptomatic side on T1 BB-FSE were higher than those on CE-GRE because of the smaller influence of susceptibility artifacts in T1 BB-FSE based on SE images. Finally, in terms of the pathophysiology of facial neuritis, if inflammation and edema occur in the labyrinthine and anterior genu segments surrounded by rigid temporal bone, it can lead to vascular compromise [5,6,7]. We can speculate that vascular compromise in these two segments is one of the reason why the rSIs of the two segments were lower than those of the canalicular segment.

The findings of this study are clinically meaningful in several ways. First, this is the first study to investigate the diagnostic ability of the T1 BB-FSE sequence for facial neuritis. Second, this study revealed that the T1 BB-FSE image has superior performance in terms of sensitivity, specificity, and accuracy compared to the previous sequence without comparison with pre-contrast sequences or the use of the subtraction method. Third, in terms of the interrater agreement for diagnostic confidence, the results were better than those of the preexisting sequence. Fourth, since there are various pathological conditions associated with facial nerves such as inflammatory diseases other than facial neuritis, leptomeningeal metastasis, and perineural tumor spread that can also show contrast enhancement of facial nerve on MRI [19], knowing the diagnostic accuracy of contrast enhancement lesion of the facial nerve on the novel MR sequence might be helpful in clinical practice. Lastly, as a routine clinical sequence, the scan time of the T1 BB-FSE images used in this study is acceptable.

This study has several limitations, including the intrinsic limits of its retrospective design. First, pre-contrast 3D T1WI was not performed. Thus, the absolute SI increase in the facial nerve could not be compared and the subtraction method could not be used. However, in clinical practice, when performing MRI to exclude other reasons in facial neuritis patients with atypical manifestations, pre-contrast 3D T1WI is usually not routinely used because of the limit of the overall MR acquisition time to include other multiple sequences. Second, we kept the scan orders of the two sequences consistent. Therefore, since the accumulated amount of gadolinium in the inflamed facial nerve changes as the scan time elapses, the possibility of an error when the signal intensity of the sequence is performed first is smaller than that of the sequence performed later but cannot be excluded. However, according to a previous study, it is known that the time needed for enhancement of the contrast in the 3D FSPGR sequence to reach its peak is approximately 1 min after gadolinium injection, and the maximum concentration lasts approximately 6 min [44]. In our study, the scan delays of the CE-GRE and T1 BB-FSE sequences after gadolinium injection were 2 min and 7 min, respectively. Therefore, the possibility of errors due to the scan order is expected to be small. Third, the scan time from symptom onset was between 0 and 29 days, and it is possible that this time variability influenced the LEC and rSI values. Fourth, the extent of the contrast-enhancing lesion was not measured in the labyrinthine segment. However, because the labyrinthine segment is the shortest segment of the facial nerve [45], it is too short to measure the extent of contrast enhancement accurately; thus, there is a potential risk that small changes in the measurements might be largely reflected. Lastly, we did not investigate the clinical significance of the T1 BB-FSE sequence, including the correlation with severity of symptoms, electrophysiological studies, and prognosis of patients with facial neuritis. However, the purpose of this study was primarily to determine the diagnostic ability of the T1 BB-FSE sequence in facial neuritis.

## 5. Conclusions

Despite some limitations, this study revealed that the T1 BB-FSE sequence could significantly improve the diagnostic ability for facial neuritis compared with the preexisting CE-GRE sequence. These results suggest that the T1 BB-FSE sequence can be used as a routine clinical sequence in the initial evaluation of facial neuritis based on the promising data and acceptable scan time. Future studies in a large patient population are needed to elucidate the correlation between the T1 BB-FSE sequence and the severity of the symptoms and prognosis of facial neuritis.

## Figures and Tables

**Figure 1 jcm-10-01850-f001:**
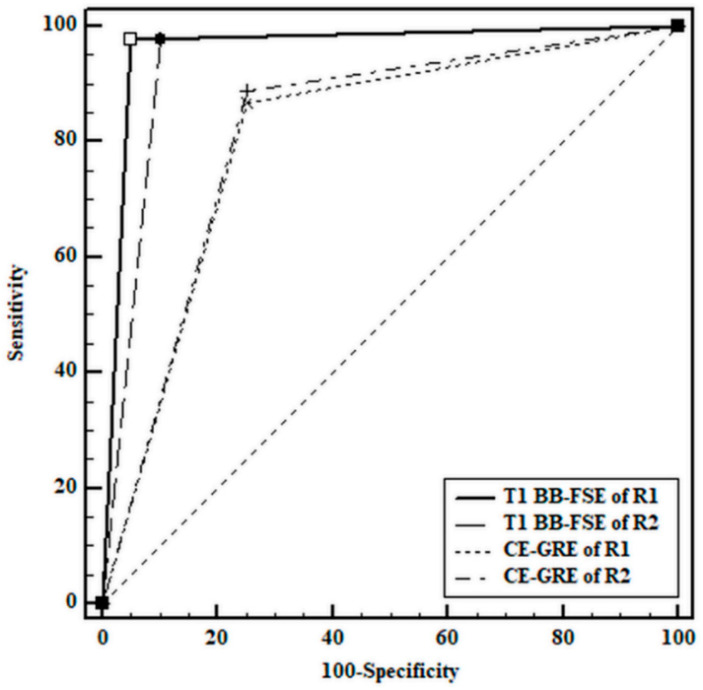
Receiver operating characteristic curves for the diagnosis of facial neuritis. R1 = reviewer 1; R2 = reviewer 2.

**Figure 2 jcm-10-01850-f002:**
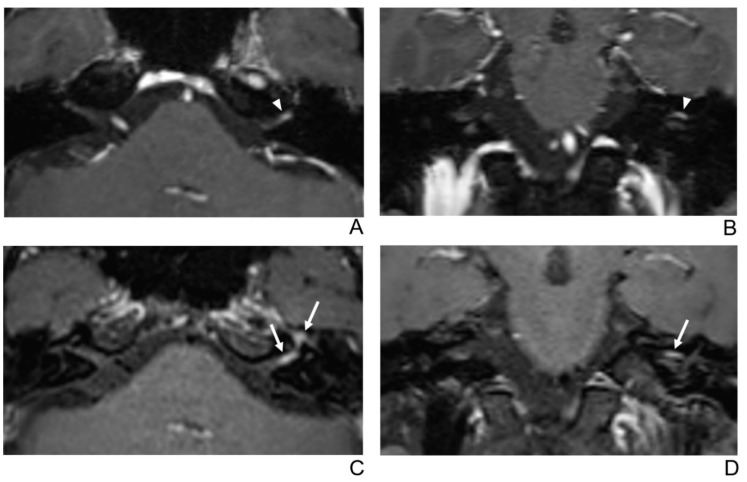
A 39-year-old male patient with left facial neuritis. (**A**–**D**) Paired axial and coronal CE-GRE (**A**,**B**) and T1 BB-FSE (**C**,**D**) images of the same patient. The left distal canalicular segment of the facial nerve showed asymmetric, intense enhancement in the axial (**A**) and coronal (**B**) CE-GRE images (arrowheads). In contrast, the labyrinthine segment showed no definite enhancement, and the anterior genu segment showed moderate degree enhancement and was interpreted by reviewers 1 and 2 as left facial neuritis (2, diagnostic confidence; 3, 0, and 2, visual grades for contrast enhancement (CE) in the canalicular, labyrinthine, and anterior genu segments, respectively). The left facial nerve shows asymmetric intense enhancement in the canalicular, labyrinthine, and anterior genu segments in the axial (**C**) and coronal (**D**) T1 BB-FSE images (arrows); it was evaluated by reviewers 1 and 2 as left facial neuritis (2, diagnostic confidence; 3, 2, and 3, visual grades for CE in the canalicular, labyrinthine, and anterior genu segments, respectively). Compared to the CE-GRE images shown earlier, in the T1 BB-FSE image, it can be seen that the contrast-enhanced area in each segment of the facial nerve is longer and wider.

**Figure 3 jcm-10-01850-f003:**
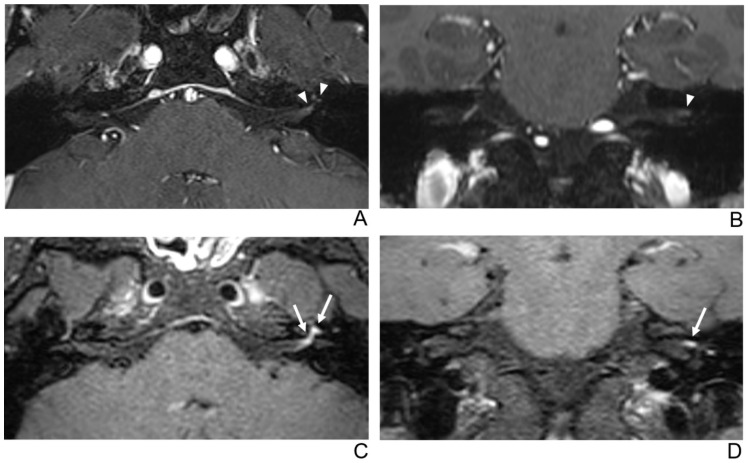
A 29-year-old male patient with left facial neuritis. (**A**–**D**) Paired axial and coronal CE-GRE (**A**,**B**) and T1 BB-FSE (**C**,**D**) images of the same patient. The left distal canalicular segment of the facial nerve showed faint enhancement in the axial (**A**) and coronal (**B**) CE-GRE images (arrowheads). Those images were interpreted by reviewers 1 and 2 as suspicious left facial neuritis (1, diagnostic confidence; 1, 0, and 1, visual grades for contrast enhancement (CE) in the canalicular, labyrinthine, and anterior genu segments, respectively). In contrast, the left facial nerve shows asymmetric, intense enhancement in the canalicular, labyrinthine, and anterior genu segments in the axial (**C**) and coronal (**D**) T1 BB-FSE images (arrows); it was evaluated by reviewers 1 and 2 as left facial neuritis (2, diagnostic confidence; 3, visual grade for CE in all three segments).

**Figure 4 jcm-10-01850-f004:**
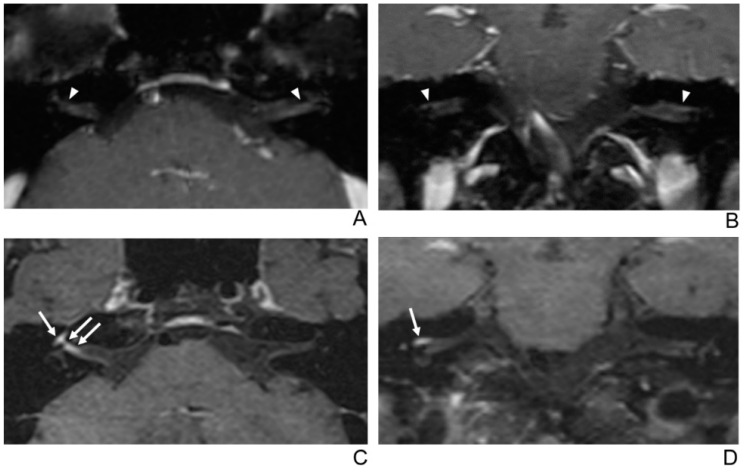
A 56-year-old female patient with right facial neuritis. (**A**–**D**) Paired axial and coronal CE-GRE (**A**,**B**) and T1 BB-FSE (**C**,**D**) images of the same patient. In the axial (**A**) and coronal (**B**) CE-GRE images, no definite asymmetric enhancement was demonstrated in the bilateral facial nerves (arrowheads); thus, it was evaluated by reviewers 1 and 2 as normal facial nerves (false negative). In contrast, the right facial nerve shows asymmetric intense enhancement in the canalicular, labyrinthine, and anterior genu segments in the T1 BB-FSE axial (**C**) and coronal (**D**) images (arrows); thus, the two reviewers interpreted the enhancement as right facial neuritis (true positive; 2, diagnostic confidence; 3, 2, and 3, visual grades for CE in the canalicular, labyrinthine, and anterior genu segments, respectively).

**Figure 5 jcm-10-01850-f005:**
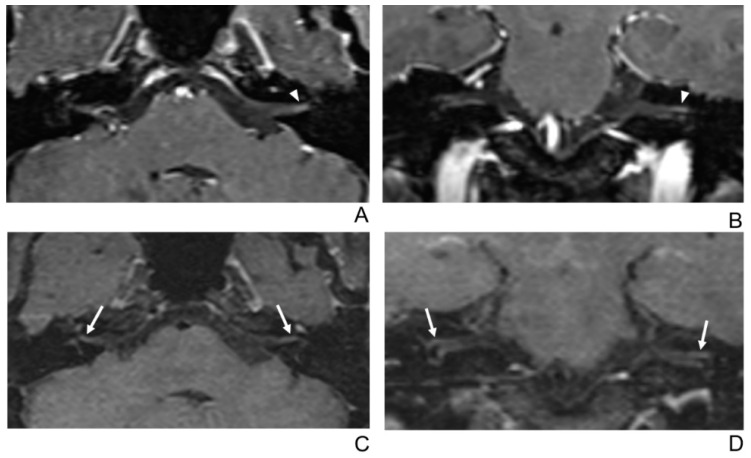
A 68-year-old female without symptoms of facial palsy (normal control). (**A**–**D**) Paired axial and coronal CE-GRE (**A**,**B**) and T1 BB-FSE (**C**,**D**) images of bilateral facial nerves. The only distal canalicular segment of the left facial nerve demonstrates moderate asymmetric enhancement in the axial (**A**) and coronal (**B**) CE-GRE images (arrowheads); thus, each reviewer interpreted it as left facial neuritis (false positive; 1, diagnostic confidence; 2, visual grade for contrast enhancement in the canalicular segment). In contrast, the bilateral facial nerves showed no significantly asymmetric enhancement in the T1 BB-FSE axial (**C**) and coronal (**D**) images (arrows); two reviewers evaluated it as normal facial nerves (true negative).

**Table 1 jcm-10-01850-t001:** Baseline characteristics of the study population ^a^.

Clinical Characteristics	Facial Neuritis (*n* = 45)
Mean age (yrs.)	49.6 (25–80)
Sex (Male/Female)	26/19
Affected side (Rt./Lt.)	26/19
Mean duration time (days)	7.1 (0–29)

^a^ The data represent the number of patients, unless otherwise noted.

**Table 2 jcm-10-01850-t002:** Results of diagnostic accuracy of the two reviewers ^a^.

	Reviewer 1		*p* ^a^	Reviewer 2		*p* ^a^	Total		*p* ^a^
	T1-BB-FSE	CE-GRE		T1-BB-FSE	CE-GRE		T1-BB-FSE	CE-GRE	
Sensitivity (%) (TP/Disease)	97.8 (44/45)	86.7 (39/45)	0.063	97.8 (44/45)	88.9 (40/45)	0.125	97.8 (88/90)	87.8 (79/90)	0.004
Specificity (%) (TN/Normal)	95 (19/20)	75 (15/20)	0.219	90 (18/20)	75 (15/20)	0.375	92.5 (37/40)	75 (30/40)	0.065
Accuracy (%)	96.9 (63/65)	83.1 (54/65)	0.012	95.4 (62/65)	84.6 (55/65)	0.039	96.2 (125/130)	83.8 (109/130)	<0.001
PPV (%)	97.8	88.6		95.7	88.9				
NPV (%)	95	71.4		94.7	75				
AUC	0.964 (0.885–0.994)	0.808 (0.692–0.895)	0.014	0.939 (0.850–0.983)	0.819 (0.704–0.904)	0.042			

^a^ These results are compared using McNemar’s test for sensitivity, specificity, and accuracy and DeLong’s test for AUCs. Note: TP and TN: evaluated by reviewers as facial neuritis and normal, respectively; PPV: positive predictive value; NPV: negative predictive value. Disease: patient with facial neuritis.

**Table 3 jcm-10-01850-t003:** Results of the qualitative analysis ^a^.

Assessment	T1 BB-FSE	Agreement, *n* (%)	CE-GRE	Agreement, *n* (%)	*p* (Intra-Reviewer Comparison)
Diagnostic confidence		38 (84%)		37 (82%)	
Reviewer 1	1.87 ± 0.06 (1.75–1.99)		1.49 ± 0.73 (1.27–1.71)		0.002
Reviewer 2	1.80 ± 0.46 (1.66–1.94)		1.49 ± 0.69 (1.28–1.70)		0.002
Mean	1.83 ± 0.43 (1.74–1.92)		1.49 ± 0.71 (1.34–1.64)		<0.001
Visual asymmetric enhancement					
Canalicular		33 (73%)		32 (71%)	
Reviewer 1	2.47 ± 0.76 (2.24–2.69)		1.82 ± 0.89 (1.56–2.09)		<0.001
Reviewer 2	2.51 ± 0.66 (2.31–2.71)		2.02 ± 0.66 (1.82–2.22)		<0.001
Mean	2.49 ± 0.71 (2.34–2.64)		1.92 ± 0.78 (1.76–2.09)		<0.001
Labyrinthine		33 (73%)		31 (69%)	
Reviewer 1	1.60 ± 0.75 (1.37–1.83)		1.20 ± 0.73 (0.98–1.42)		0.005
Reviewer 2	1.71 ± 0.69 (1.50–1.92)		1.04 ± 0.88 (0.78–1.31)		<0.001
Mean	1.66 ± 0.72 (1.50–1.81)		1.12 ± 0.80 (0.95–1.29)		<0.001
Anterior genu		37(82%)		33 (73%)	
Reviewer 1	0.84 ± 0.52 (0.69–1.00)		0.87 ± 0.84 (0.61–1.12)		0.979
Reviewer 2	0.71 ± 0.46 (0.57–0.85)		0.71 ± 0.97 (0.42–1.00)		0.900
Mean	0.78 ± 0.49 (0.67–0.88)		0.79 ± 0.91 (0.60–0.98)		0.852

^a^ The mean values of the two reviewers are reported with standard deviations and 95% confidence intervals.

**Table 4 jcm-10-01850-t004:** Quantitative measurements of the LEC and rSI of each segment (mean values of three time measurements).

	T1-BB-FSE	CE-GRE	*p*-Value ^a^
Enhancing lesion extent			
Length (median, (IQR))	4.08 mm, (3.31–4.67)	2.23 mm, (1.80–3.44)	<0.001
Width (median, (IQR))	1.60 mm, (1.41–1.87)	1.42 mm, (1.17–1.72)	0.026
rSI of each segment			
Canalicular (median, (IQR))	2.00 (1.68–2.31)	1.73 (1.29–2.17)	0.029
Labyrinthine (median, (IQR))	1.84 (1.50–2.50)	1.67 (1.40–2.12)	0.193
Anterior genu (median, (IQR))	1.38 (1.24–1.70)	1.25 (1.13–1.85)	0.480

^a^ These measurements were compared by the Mann–Whitney test. Note: IQR: interquartile range; rSI: affected side-to-normal signal intensity ratio.

## Data Availability

The data presented in this study are available upon request from the corresponding author. The data are not publicly available, as participants of this study did not agree for their data to be shared publicly.

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
