# Peer review of "Improvement of the Diagnostic Performance of Facial Neuritis Using Contrast-Enhanced 3D T1 Black-Blood Imaging: Comparison with Contrast-Enhanced 3D T1-Spoiled Gradient-Echo Imaging"

_jcm, 2021, doi:10.3390/jcm10091850_

Round 1

Reviewer 1 Report

This is an excellent article for anyone working on facial palsy. The quality of the writing is clear and fluid. The methodology used is scientifically very rigorous. 

Some remarks:
- According to the latest American recommendations, MRI should be performed systematically for all facial palsies.
- I find unfortunate that the reviewers were blinded to the pathological side. It is certainly clever, but through the eye of a clinician, it has no clinical logic since we all know the affected side when examining the patient. Indeed, MRI has strictly no interest for the positive diagnosis of facial palsy. It is useful to appreciate the severity (related to the etiology, and if possible the prognosis). So, I wish the authors would be aware of this and write it down.
- Also, I would have liked a more thorough reflection on the pathophysiology of Bell's Palsy, especially by the mechanisms of compression / edema / ischemia that occur in the narrowest part of the bony canal, i.e. the labyrinthine portion. Hervochon in 2019 wrote about this by studying the shape of the bony canal.

In spite of the 3 problems raised above and the lack of data on the prognosis of recovery of these paralysis, I believe that this work totally deserves to be published, since it brings a concrete change in the management of patients by modifying the MRI protocols.

Author Response

All the authors greatly appreciate helpful and warm comments of reviewer 1.

Please see the attachment file entitled "Response letter to Reviewer 1's comments"

Thank you again for giving us the opportunity to revise and resubmit this manuscript.

Sincerely

- Dr SWJ 

Reviewer 2 Report

The study aims to investigate the diagnostic ability of T1 BB-FSE sequence compared with the CE-GRE sequence in the assessment of patients with facial neuritis.

The methods are well designed as both subjective and quantitative assessment were performed. 

Interestingly the T1 BB-FSE sequence performed better than the CE-GRE sequence. This finding has never been reported with regards with Bell's palsy. 

INTRODUCTION

"Facial neuritis (Bell's palsy) is the most common cause of acute periph- eral facial nerve palsy in adults, and patients usually show unilateral facial nerve palsy without a clear cause within 72 hours"

This sentence is not clear. Please rephrase. 

"This disease is diagnosed primarily on the basis of characteristic clinical symptoms, physical examination and electrodiagnostics such as electroneurography or electromyography."

Bell's palsy is diagnosed after a thorough clinical assessment and radiologic study, primarily to exclude possible differential diagnosis. The role of electroneurography and electromyography is limited to the assessment of the prognosis during the early onset and later during the follow-up. 

 METHODS

The radiologic study protocol is well explained. 

Statistical methods are sound and adequate.

It is important to specify that these patients were followed up and studied with complete clinical, audiologic and radiologic assessment to rule out other possible differential diagnosis (e.g. parotid plan neoplasms, facial nerve neoplasms, cholesteatoma, etc.).

It is of utmost importance to avoid misdiagnosis of Bell's Palsy whose diagnosis can be confirmed only exclusion of other differential diagnosis.

How late was the MRI performed after the facial palsy onset? Probably it would be correct to define in the inclusion criteria this aspect, otherwise the evaluation of facial nerve enhancement can.

RESULTS

"The sensitivity (97.2%) and accuracy (96.2%) of T1 BB-FSE were signifi- cantly higher than those of CE-GRE (sensitivity, p = 0.004; accuracy, p < 0.001)."

Please specify the sensitivity and accuracy of CE-GRE.

DISCUSSION 

It is well written and organised. 

CONCLUSION

Good

Author Response

All the authors greatly appreciate helpful and warm comments of reviewer 2.

Please see the attachment file entitled "Response letter to Reviewer 2's comments"

Thank you again for giving us the opportunity to revise and resubmit this manuscript.

Sincerely

- Dr SWJ
